# Roles of Telomere Biology in Cell Senescence, Replicative and Chronological Ageing

**DOI:** 10.3390/cells8010054

**Published:** 2019-01-15

**Authors:** Jun Liu, Lihui Wang, Zhiguo Wang, Jun-Ping Liu

**Affiliations:** 1Institute of Ageing Research, School of Medicine, Hangzhou Normal University, Hangzhou 311121, Zhejiang, China; wanglihui@hznu.edu.cn (L.W.); zhgwang@hznu.edu.cn (Z.W.); 2Department of Immunology, Monash University Faculty of Medicine, Melbourne, Vitoria 3004, Australia; 3Hudson Institute of Medical Research, Clayton, Victoria 3168, Australia; 4Department of Molecular and Translational Science, Monash University, Clayton, Victoria 3168, Australia

**Keywords:** telomere, ageing, lifespan, senescence, replicative ageing, chronological ageing, yeast

## Abstract

Telomeres with G-rich repetitive DNA and particular proteins as special heterochromatin structures at the termini of eukaryotic chromosomes are tightly maintained to safeguard genetic integrity and functionality. Telomerase as a specialized reverse transcriptase uses its intrinsic RNA template to lengthen telomeric G-rich strand in yeast and human cells. Cells sense telomere length shortening and respond with cell cycle arrest at a certain size of telomeres referring to the “Hayflick limit.” In addition to regulating the cell replicative senescence, telomere biology plays a fundamental role in regulating the chronological post-mitotic cell ageing. In this review, we summarize the current understandings of telomere regulation of cell replicative and chronological ageing in the pioneer model system *Saccharomyces cerevisiae* and provide an overview on telomere regulation of animal lifespans. We focus on the mechanisms of survivals by telomere elongation, DNA damage response and environmental factors in the absence of telomerase maintenance of telomeres in the yeast and mammals.

## 1. Telomere Structure and Maintenance

Telomere contains telomeric DNA and its binding proteins. The telomeric DNA sequence is TG_1–3_, terminated with a 3′end G-rich overhang, synthesized by telomerase that consists of the catalytic subunit Est2 [1], the RNA template Tlc1 [2], two regulatory subunits—Est1 [3] and Est3 [4], the Yku80 subunit [5] and the telomeric single-stranded DNA binding protein—Cdc13 [6], Pop1, Pop6 and Pop7 [7] in the budding yeast. In addition, the telomere binding protein complex “shelterin-like telosome” consists of telomeric duplex-region binding protein Rap1 that recruits Rif1, Rif2, Sir3 and Sir4 to telomeres via the carboxyl domains to form Rap1-Rif1/2 and Rap1-Sir3/4 capping complexes at telomeres. Cdc13 is also an integral component of the CST (Cdc13-Stn1-Ten1) complex that is conserved from yeast to human for single-strand telomere capping and regulation of the C-strand telomeres [8,9]. Telomeres cannot be fully replicated by DNA polymerases due to the “end replication” problem [10,11,12]. Most eukaryotes evolve to maintain their telomeres with specific machineries including the reverse transcriptase telomerase that employs an intrinsic RNA template to reverse transcribe the G-rich nucleotide sequence to the 3′ ends of the G-rich strand of telomeres. Several reviews have published elsewhere on yeast telomere biology [13,14] and telomere length regulation [15].

In human and mouse cells, telomere DNA sequence contains (TTAGGG)_n_ repeats and telomerase consists of the catalytic subunit TERT, RNA component TERC, dyskerin (DKC), nucleolar protein 10 (NOP10), non-histone protein 2 (NHP2), GAR1 (encoding H/ACA ribonucleoprotein complex subunit 1) and telomerase Cajal body protein 1 (TCAB1). The shelterin complex contains the duplex-region binding proteins telomeric repeat binding factor 1 (TRF1) and TRF2 that recruit other four components: protection of telomeres 1 (POT1), repressor/activator protein 1 (RAP1), TRF1-interacting nuclear factor 2 (TIN2) and TPP1 (adrenocortical dysplasia protein homologue, ACD). These subunits form a complex called shelterin to coat and cap telomeric DNA [16,17], deficiency any of which may cause telomere dysfunction and cell senescence as for the case of POT1 that is a single-stranded telomeric DNA binding protein involved in ATR-dependent DNA damage response [18]. TRF2 and TPP1/POT1 inhibit two distinct telomere-threatening 5’ end-resection pathways that are differentially regulated by the ATM (yeast Tel1 orthologue) and ATR (yeast Mec1 orthologue) DNA damage signaling kinases [19]. One pathway is ATM-activated CtIP/MRN (Mre11-Rad50-Nbs1)-mediated limited resection, which is inhibited by TRF2; the other is ATR-stimulated Exo1/BLM-mediated extensive resection which is inhibited by TPP1/POT1 [19]. Both 5′ end-resection pathways are repressed by 53BP1 and hRif1, which are all stimulated by ATM and ATR kinases that are suppressed by shelterin complex [19].

## 2. Cell Replicative Ageing versus Cell Replicative Senescence

Ageing is defined as age-dependent functional decline with gradual losses of reproductivity, which is determined by multiple genetic and environmental factors. Replicative ageing represents reproductive ageing in budding yeast, referring to the time-dependent decline of the capacity of a mother cell to produce its daughter cells [20]. A mystery if a cell divides unendingly to produce its daughter cells was unveiled in 1959. Mortimer and Johnston reported for the first time that single baker’s yeast cell produces only a fixed number of daughter cells by asymmetrical budding before the mother cells enter a senescent state and lyse, with the limited budding potential named replicative (reproductive) lifespan [20]. This study demonstrated that instead of chronological time passage, cell budding results in replicative ageing in the single-cell eukaryotic organism.

Since telomerase is expressed in wild-type yeast cells, replicative ageing appears not to be set off by critically short telomeres, potentially allowing quantitative analysis of cell replicative lifespan in the yeast *Saccharomyces cerevisiae* model system to identify genes and chemical compounds in the regulation of cell replicative ageing [21,22,23,24,25,26]. On the other hand, the concept of cell replicative senescence (or cell senescence hereafter) refers to the state of permanent cell cycle arrest caused by consecutive symmetrical cell duplications, critically short telomeres and DNA damage response in yeasts and mammals [3,27]. However, cells with critically short telomeres are able to evade senescence by lengthening their telomeres via amplification of the subtelomeric Y’ elements [28] and homologous recombination between the telomere-end heterogeneous TG_1–3_ sequences [29].

In human somatic diploid cells, Leonard Hayflick and his colleagues reported in early 1960s that cultured fibroblasts become aged with limited cell divisions [30,31]. This is because human normal somatic diploid cells do not have significant telomerase activity and fail to maintain their short telomeres so that cells enter a permanent cell cycle arrest. The notion of “Hayflick limit” denotes that somatic cells divide a fixed number of times, with human cells such as fibroblasts dividing forty to sixty times, before cell senescence [30,31,32].

In the budding yeast *Saccharomyces cerevisiae*, measuring cell senescence was invaluable to identify the genes that are essential for telomerase recruitment and assembly [1,3,4]. Two types of yeast survivors evaded from senescence with critically short telomeres have been discovered [28,29] (see below). However, there has been relatively scarce evidence yet to demonstrate molecular interactions between the pathways that lead to cell replicative senescence (caused by critically short telomeres) and replicative (reproductive) ageing independent of telomere length.

## 3. Cell Chronological Ageing

A chronological ageing model was reported first by Longo and colleagues in the model system *Saccharomyces cerevisiae*, showing that non-dividing yeast cells lose their viabilities in a manner depending on both nutrient sufficiency and age, in order to model post-mitotic ageing of human cells such as neurons and muscle cells [33,34]. Chronological lifespan is usually defined as the time period when cells remain viable at stationary phase (G_0_ phase) [33]. The budding yeast post-mitotic cell ageing usually undergoes for 1–2 months of survival and has been widely used in ageing research [26,35,36,37,38,39,40,41,42]. It is believed that evolutionarily conserved key molecules in the polarization processes from yeast budding and mating to metazoan neuronal outgrowth and spinogenesis are homologous, potentially making the yeast chronological ageing process as a useful model for the non-dividing human cell ageing research [43,44,45,46,47,48,49].

## 4. Roles of Telomere Length in Cell Senescence, Replicative and Chronological Ageing

It has become evident that critically short telomeres in the absence of telomerase cause cell senescence, while reintroduction of telomerase rescues senescence [3,50], indicating a causal relationship between critically short telomeres and cell senescence [51]. The first telomerase gene identified was in yeast, named *EST1* (ever shorter telomeres) [3]. Cells with *EST1* gene knock-out are not immediately unviable but rather senesce following successive passages with telomeres gradually shortened to critically short length [3]. These studies show that when telomeres are critically short, cell senescence mechanisms are activated to drive cells into a permanent cell cycle arrest. Reintroduction of telomerase to the cells null of telomerase increases the replicative lifespan, indicating a pivotal role of telomere length above the critically short point in cell replicative lifespan [50,52,53,54].

However, it has been shown that inappropriately prolonged telomeres shorten budding yeast replicative lifespan, whereas significantly shorter-than-normal telomere length due to telomerase deficiency extends yeast replicative lifespan [55]. Consistently, preventing telomere lengthening by inhibiting telomere recombination promotes yeast replicative lifespan extension [56]. Why is the lifespan extended in the strain with shorter telomeres? Mechanistic studies show that the yeast chromatin silencing machinery, encoded by *SIR2*, *SIR3* and *SIR4*, undergoes redistribution from telomeres to non-telomere sites when telomeres are shortened to increase heterochromatin maintenance, genome stability and the lifespan, and deletion of either *SIR3* or *SIR4* decreases the lifespan [55]. More recently, no effect of long telomeres on vegetative cell division, meiosis or in cell chronological lifespan is observed in the yeast [57]. During chronological ageing, longer telomeres remain stable albeit without affecting chronological lifespan [42]. These strains with 2–4 folds longer telomeres do not carry any plasmids or gene deletions, potentially applicable to assess the relationship between overlong telomeres and chronological lifespan [42]. It thus appears that neither replicative nor chronological lifespan benefits from longer-than-normal telomeres.

## 5. Role of Telomere Shortening in Multicellular Organismal Ageing

Ageing of multicellular organisms is more complex than single eukaryotic cell organism. Telomere lengthening by activating telomerase increases longevity in mice with [58] or without risking tumorigenesis [59,60] and extends replicative lifespan in human cells [50,53,54]. Telomeres longer than normal are associated with diminished age-related pathology in humans [61]. In the nematode *Caenorhabditis elegans*, long telomeres associated with overexpression of *hnRNP-1* (encoding heterogeneous nuclear ribonucleoprotein A1) are correlated with lengthened organismal lifespan [62]. On the other hand, telomeres longer than normal are associated with increased risks of vascular hypertension [63,64] and lung adenocarcinoma [58,65].

Interestingly, it is not only telomere DNA damage response but also glucose homeostasis and inflammation that mediate the lifespan changes inflicted by altered telomere lengths in mammals. Telomerase catalytic subunit TERT binds cell membrane glucose transporter to enhance glucose import; inhibition of TERT halves glucose intake but overexpressing TERT triples the uptake [66] and glucose-enriched substitution feeding extends the short lifespan by 20% of the mice deficient of telomerase RNA subunit [67]. These are consistent with the notion that glucose homeostasis and energy sufficiency are fundamental in lifespan regulation in the maintenance of short lifespan associated with telomerase deficiency and telomere dysfunction. It is noteworthy that increased glycolysis extends fish lifespan by inhibiting polycomb repressive complexes (PRCs)-mediated H3K27me3 or expressing genes for glycolysis [68].

Recently, we showed that senescence-associated low grade inflammation (SALI) is involved in the settings of telomere dysfunction and shortening in ageing [69]. Deficiency of either TERT or TERC results in telomere dysfunction and shortening, SALI, losses of tissue stem cells and short lifespan in mice [69]. The chronic sterile SALI appears to cause telomere dysfunction in various tissues of spleen, colon and liver in addition pulmonary epithelia in mice [69]. In addition to a causal role of telomeric DNA injury to inflammation, as parts of the vicious cycle between inflammation and telomeric DNA injury, inflammatory cytokine TGF-β inhibits telomerase gene expression [70,71]. Moreover, telomere shortening induces interferon-β (IFN-β) signaling and the increased IFN-β is required in premature ageing and death induced by TERC deficiency with the mechanism downstream of telomeric DNA damage response [72]. Furthermore, inhibition of IFN-β dramatically rescuing the short lifespan imposed upon by telomerase deficiency [72].

During telomeric DNA damage response, both p16 and p21 play critical role in arresting the cell cycle temporarily for DNA damage repairs or permanently to render cell replicative senescence if damages are irreparable. More interestingly, studies demonstrate that p16 and p21 play distinct roles in regulating the lifespan of mice carrying telomeric DNA damages, with predominantly single-stranded ATR-dependent telomere DNA damage induced by deficiency of POT1 component of shelterin [18]. Deficiency of p16 drastically shortens the lifespans of mice with deficiency of POT1 or both POT1 and telomerase RNA subunit, suggesting that p16 is required for survival with telomeric DNA dysfunction [18]. In a stark contrast, however, p21 deficit remarkably extends the lifespan of mice with POT1 deficiency, without triggering ATR-dependent DNA damage response and genome instability, suggesting that p21 is required for DNA damages and short lifespan in telomere dysfunction [18].

In the mice, although telomere shortens ~100 times faster than that in humans, it is the rate of increase in the percentage of short telomeres, rather than the rate of telomere shortening per month, that matters in predicting lifespan [73]. The maximal lifespan of different mammals differs significantly by more than 100 folds, ranging from about 2 years in shrews to >200 years in bowhead whales [74]. Telomeres shorten as a function of ages in old animals, including humans with shorter telomeres in elder people (>60 years) being a prognosis for higher mortality [75]. In the long lived birds and mammals, telomeres shorten more slowly [76]. In the longevity naked and the blind mole rats that are two subterranean rodent species, telomerase is activated in somatic tissues that lack replicative senescence and resist tumorigenesis [77,78]. These studies confirm that telomere maintenance contributes to the regulation of lifespan and longevity significantly.

## 6. Telomere Maintenance by Recombination, Genome Instability and Cell Replicative and Chronological Ageing

In the yeast *Saccharomyces cerevisiae*, cells deficient of telomerase evade cellular senescence triggered with critically short telomeres to survive by either amplification of subtelomeric Y’ elements (type I survivor) [28] or telomere-end TG_1–3_ sequence recombination through homologous recombination (type II survivor) [29] (Figure 1). However, the telomerase-null post-senescent type II survivors with enhanced telomere recombination exhibit accelerated cellular replicative ageing [52]. In the absence of telomerase activity for a short term (about 25–30 generations post telomerase inactivation), the cells experience a detectable genomic instability [79]. In response to transient DNA replication stress, altered cell-cycle dynamics occurs even in young mother cells with accelerated replicative ageing independent of telomere length [79]. This premature onset of cell senescence induced by telomerase deficiency and subsequent accelerated replicative ageing induced by telomere homologous recombination may reflect telomere uncapping events in the mother cells, which requires further investigation. 

The mechanisms by which the subtelomeric Y’ elements get amplified to underpin the type I survivors remain largely elusive. Whether or not specific telomeric proteins play a role in coordinately regulated telomere homeostasis requires investigation. Sir2, Sir3 and Sir4 along with Rif1 and Rif2 are localized to duplex telomeres by binding to the C-termini of Rap1 [13,14]. Whereas Rif1 and Rif2 are negative regulators of telomerase recruitment to telomeres [80], Sir4 recruits Sir2 and Sir3 to participate in chromatin silencing [81] by interacting with not only Rap1 but also Yku80 [82,83]. While binding of Yku80/70 to telomerase TLC1 and telomeric DNA is mutually exclusive to each other [84], Sir4 is required to mediate the Yku80-TLC1 recruitment to telomeres [85,86]. Thus, it is tempting to hypothesize that in the absence of telomerase recruitment, Sir2, Sir3 and Sir4 along with Rif1 and Rif2 are involved in regulating the type II survivor lifespan.

Constitutive telomerase activity is observed by experimentally tethering the telomerase catalytic subunit Est2 to the single-stranded DNA binding protein Cdc13 with ectopic expression of the fusion gene *CDC13-EST2* [42,87]. However, the constitutively increased telomerase activity results in shortened chronological lifespan [42]. *CDC13-EST2* ectopic expression culminates in 2–4 folds longer telomeres but with significant genome instability including accumulated extra chromosomal rDNA circle species (ERC), frequent *CAN1* marker gene mutations and gross chromosomal rearrangement (GCR) in an age-dependent manner [42]. Moreover, the significantly increased genome instability issuppressed by down-regulation of the evolutionarily conserved TORC1 (the target of rapamycin complex 1)-Sch9 pathway [42], suggesting that pharmaceutical strategies (e.g., rapamycin and caffeine [88]) are potentially applicable to stabilize the genome instability caused by constitutive active telomerase to increase lifespan.

## 7. Perspectives

Telomeres are lengthened by mechanisms including telomerase-mediated reverse transcription, subtelomeric DNA amplification and telomeric DNA homologous recombination. Uncontrolled telomere lengthening such as end-to-end joining is by and large at the expenses of risking premature ageing and related disorder such as cancer. Moreover, the mechanisms that determine the different aspects of longevity, premature replicative senescence and chronological ageing appear to hinge on the extensity and intensity of telomeric DNA abnormality and on the fidelity of DNA damage response and repair. Investigations of effective and sufficient regulation of telomeric DNA damage response and repair, by examining the molecular interplays between the different pathways of telomere maintenance for example, will shed light on the mechanisms of preventing telomere damages by environmental stresses (e.g., ultraviolet radiation and oxidative stress) in ageing. Thus, elucidation of the molecular functions, the structures conferring the specific functions of telomere proteins and how environmental stress impacts on the proteostasis of telomere proteins will inform on the means to intercept stress-induced organismal premature ageing for longevity.

## Figures and Tables

**Figure 1 cells-08-00054-f001:**
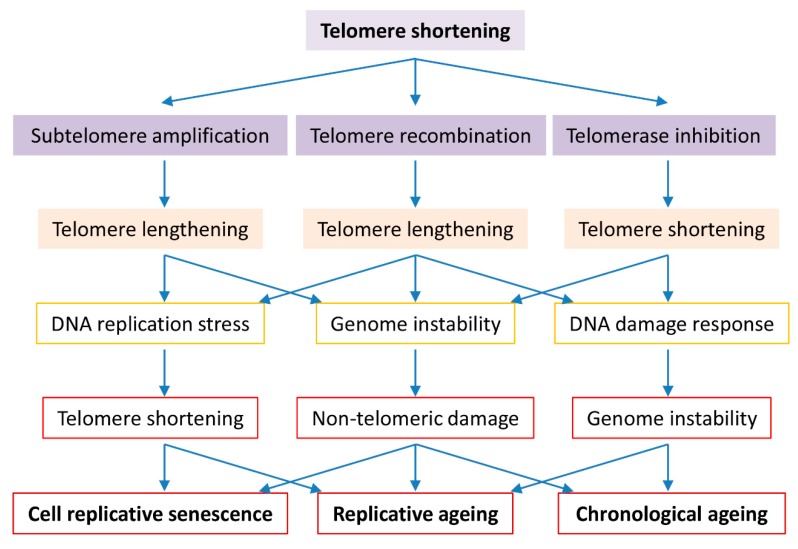
Roles of telomere biology in cell senescence, replicative and chronological ageing. Cells evolve to have regulated telomerase activity to preserve telomere homeostasis which is vital to genome stability in organism ageing. Short telomeres are maintained by telomerase in the early stage, telomere DNA homologous recombination or subtelomeric DNA amplification in the late stage of telomere shortening. Telomerase inactivation results in critically short telomeres that either activate the cell cycle check-point resulting in cell senescence or promote telomere-telomere recombination or subtelomere amplification resulting in genome instability and replicative difficulty. Long telomere associated genome instability and replicative difficulty cause not only telomere shortening but also non-telomeric damages, culminating in cell senescence, replicative ageing and accelerated chronological ageing.

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
