# Peer review of "Roles of Telomere Biology in Cell Senescence, Replicative and Chronological Ageing"

_cells, 2019, doi:10.3390/cells8010054_

Round 1

Reviewer 1 Report

In a thought-provoking review article, Liu et al. elegantly present and judiciously discuss recent progress in comprehending molecular mechanisms through which telomere structure and maintenance influence the replicative and chronological modes of aging in the budding yeast Saccharomyces cerevisiae. This well organized and clearly written manuscript is a must-read for anyone interested in understanding how telomere homeostasis defines the pace of replicative and chronological of aging in this unicellular eukaryote. As such, the manuscript provides a significant new contribution to the field of cellular aging. I therefore strongly recommend accepting it for publication.

Author Response

Response to Reviewer 1 Comments

In a thought-provoking review article, Liu et al. elegantly present and judiciously discuss recent progress in comprehending molecular mechanisms through which telomere structure and maintenance influence the replicative and chronological modes of aging in the budding yeast Saccharomyces cerevisiae. This well organized and clearly written manuscript is a must-read for anyone interested in understanding how telomere homeostasis defines the pace of replicative and chronological of aging in this unicellular eukaryote. As such, the manuscript provides a significant new contribution to the field of cellular aging. I therefore strongly recommend accepting it for publication.

Our response: Thank you very much for the positive comments. We have extensively revised the manuscript and hope it is now suitable for publication.

Reviewer 2 Report

The Authors of the manuscript focused on reviewing the role of telomeres in cell senescence, both replicative and chronologic. The approach to the subject is interesting. The manuscript is well written. The content bases on the recent literature. A valuable discussion concerns the clarification of terminological complexity. The only thing missing is a cartoon comparing the structure of human/mouse and yeast telomeres and a few words about the risk of cancer as a result of prolonged telomerase activity (it could be a consequence of genomic instability, what was mentioned by the Authors). The summary picture arranges the whole story very well. Some slight editorial corrections are required.

Minor comments

Chapter 3 has to be extended. Ageing of post-mitotic cells is a very hot and little recognized topic and should be thoroughly discussed. Neurons are not the only post-mitotic cells.

The second paragraph of Chapter 6 does not have any references (lines 135-146). The references are essential especially for the discussion about different stages of telomerase inactivation.

In Chapters 6 and 7 there are some unnecessary repetitions.

Line 140: “Different from the early stage after telomerase inactivation, late after telomerase inactivation results in critically short telomeres that either activates check-point and leads to senescent death….”. What have the Authors meant writing “senescent death”. Senescence does not mean that the cell will die. On the contrary, senescent cells are more resistant to apoptosis. They are alive for quite a long period of time and are metabolically active. In figure 1 the Authors have written that check-point activation causes cell cycle arrest and senescence. This scenario is better than senescent death.

Author Response

Response to Reviewer 2 Comments

The Authors of the manuscript focused on reviewing the role of telomeres in cell senescence, both replicative and chronologic. The approach to the subject is interesting. The manuscript is well written. The content bases on the recent literature. A valuable discussion concerns the clarification of terminological complexity.

Our response: Thank you very much for your positive comments. We have extensively revised the manuscript.

Point2: The only thing missing is a cartoon comparing the structure of human/mouse and yeast telomeres and a few words about the risk of cancer as a result of prolonged telomerase activity (it could be a consequence of genomic instability, what was mentioned by the Authors). The summary picture arranges the whole story very well. Some slight editorial corrections are required.

Our response: A cartoon comparing the structure of human/mouse and yeast telomeres have been shown in recent literature, e.g. Mattarocci et al., 2016 Frontiers in Genetics, so we will not try to repeat this.

Minor comments

Point3: Chapter 3 has to be extended. Ageing of post-mitotic cells is a very hot and little recognized topic and should be thoroughly discussed. Neurons are not the only post-mitotic cells.

Our response: We have extended Chapter 3 as requested.

Point4: The second paragraph of Chapter 6 does not have any references (lines 135-146). The references are essential especially for the discussion about different stages of telomerase inactivation.

Our responses: These have been corrected.

Point5: In Chapters 6 and 7 there are some unnecessary repetitions.

Our response: We have deleted unnecessary repetitions.

Point 6: Line 140: “Different from the early stage after telomerase inactivation, late after telomerase inactivation results in critically short telomeres that either activates check-point and leads to senescent death….”. What have the Authors meant writing “senescent death”. Senescence does not mean that the cell will die. On the contrary, senescent cells are more resistant to apoptosis. They are alive for quite a long period of time and are metabolically active. In figure 1 the Authors have written that check-point activation causes cell cycle arrest and senescence. This scenario is better than senescent death.

Our response: We have changed “senescent death” to senescence as suggested.

Reviewer 3 Report

In this review entitled “Roles of telomere biology in cell senescence, replicative and chronological aging”, Liu et al., attempt to give an overview of the impact of telomere biology in cell senescence and aging. The topic is very interesting. However, the authors do not provide sufficient information on the mechanisms regulating telomere stability, or even telomere length homeostasis, to understand the role of telomeres in replicative senescence. From this review it is not clear how telomere shortening can induce cell senescence or replicative aging. The authors provide correlative information on telomere length homeostasis, replicative life span and the onset of senescence. This information is often incomplete and misleading, and several sentences need to be rephrased.

Some examples:

Abstract, line 14: “telomerase as specialized reverse transcriptase adds the G-rich nucleotides to telomere ends”. This information is misleading and incorrect. The entire abstract should be revised.

Lines 26 and 27, again the sentence regarding telomerase activity at telomeres is vague and incomplete.

Lines from 88 to 98: The sentences should be rephrased, information provided is imprecise and incomplete.

Lines from 101 to 105: not clear.

Line 106: what is HRP-1?

The figure is not clear, different colors are used with no explanations and the manuscript text does not provide sufficient information to follow any of the many arrows shown.

Author Response

Response to Reviewer 3 Comments

In this review entitled “Roles of telomere biology in cell senescence, replicative and chronological aging”, Liu et al., attempt to give an overview of the impact of telomere biology in cell senescence and aging. The topic is very interesting.

Point1: However, the authors do not provide sufficient information on the mechanisms regulating telomere stability, or even telomere length homeostasis, to understand the role of telomeres in replicative senescence. From this review it is not clear how telomere shortening can induce cell senescence or replicative aging.

Our response: Thank you very much for the constructive comments. We have extensively revised the manuscript and hope it is suitable for publication. This review mainly focuses on how telomere biology affect cell senescence, replicative and chronological aging. Therefore, we will not talk too much about how telomeres are regulated.

Point2: The authors provide correlative information on telomere length homeostasis, replicative life span and the onset of senescence. This information is often incomplete and misleading, and several sentences need to be rephrased.

Some examples:

Abstract, line 14: “telomerase as specialized reverse transcriptase adds the G-rich nucleotides to telomere ends”. This information is misleading and incorrect. The entire abstract should be revised.

Our response: We have rephrased the sentence and revised the abstract as suggested.

Lines 26 and 27, again the sentence regarding telomerase activity at telomeres is vague and incomplete.

Our response: We think “telomerase activity” is appropriate.

Lines from 88 to 98: The sentences should be rephrased, information provided is imprecise and incomplete. Lines from 101 to 105: not clear.

Our response: We have rephrased some of the sentences as requested.

Line 106: what is HRP-1?

Our response: We have added explanation for HRP-1 of C. elegans.

Point 3: The figure is not clear, different colors are used with no explanations and the manuscript text does not provide sufficient information to follow any of the many arrows shown.

Our response: Figure 1 have been extensively revised.

Round 2

Reviewer 3 Report

The authors have revised their manuscript by including new parts in the text. Several issues still need to be clarified:

The authors write of a telomeric DNA damage response in various instances throughout the manuscript. At line 168, the authors even write about the “ATR-dependent telomere DNA damage induced by deficiency of Pot1”. In the first paragraph of their review the authors should describe how TRF2 and POT1 protect chromosome ends from DNA damage response.

The authors point out that in their review they will not discuss how telomeres are regulated. However, they indicate deprotected telomeres as one of the triggers of replicative senescence. Since the focus of this review is, by the title, the “role of telomere biology in cell senescence” the authors should explain what deprotected telomeres are.

The authors start the first paragraph of the review by indicating that telomeres cannot be fully replicated and discuss the activity of telomerase. How can the readers understand these concepts if the authors have not yet discussed the telomeric DNA sequence and structure? These first lines of the first paragraph should be placed at the end of the paragraph. The authors should at least mention that telomeres terminate with a G-rich 3’ overhang. 

In several instances concepts are introduced but not explained. As result the review is difficult to read.

For example, at lines 119-122, the authors write “yeast silencing machinery, encoded by SIR2, SIR3, and SIR4, undergoes redistribution from telomeres to non-telomere sites when telomeres are shortened to increase the lifespan, and deletion of either SIR3 or SIR4 decreases the lifespan [47]” How would this work? Why redistribution of Sir proteins may increase lifespan? Is the telomeric chromatin involved? Is replicative stress at telomeres involved? Are extratelomeric roles of Sir proteins involved? Could telomere transcription be involved?

Similarly, at lines 223-225: “More intriguingly, the significantly increased genome instability is suppressed by down-regulation of the evolutionarily conserved TORC1 (the target of rapamycin complex 1)-Sch9 pathway [50].” Why is this intriguing? What does this indicate?

In the perspectives section, the authors write: “The mechanisms that determine the different aspects of longevity, premature replicative senescence or chronological ageing appear to hinge on the extensity and intensity of telomeric DNA abnormality, and on the fidelity of DNA damage response and repair”. Which telomeric DNA abnormalities that impact on senescence and longevity are the authors referring to? And which DNA repair pathways? 

Still in the perspectives sections, what do the authors mean by “The different lengthening mechanisms are to different degrees at the expenses of risking premature ageing and related disorders”? or by “means to intercept stress-induced organismal premature ageing for longevity”?

The manuscript requires English editing:

some examples: "telomerase expresses in wild type yeast" line 65; "gradually loss telomere sequence" line 108

Author Response

Responses to reviewer #3 second round comments

Comments and Suggestions for Authors
The authors have revised their manuscript by including new parts in the text. Several issues still need to be clarified:

The authors write of a telomeric DNA damage response in various instances throughout the manuscript. At line 168, the authors even write about the “ATR-dependent telomere DNA damage induced by deficiency of Pot1”. In the first paragraph of their review the authors should describe how TRF2 and POT1 protect chromosome ends from DNA damage response.

Author response: Thank you for your comments. Telomeric DNA damage response is a consequence of telomere dysfunction and involved in cell aging. Accordingly, we have inserted information on POT1 in the introduction with a reference cited in line 52-55. Moreover, we have described how TRF2 and TPP1/POT1 protect chromosome ends from DNA damage response as suggested and cited appropriate publications.

The authors point out that in their review they will not discuss how telomeres are regulated. However, they indicate deprotected telomeres as one of the triggers of replicative senescence. Since the focus of this review is, by the title, the “role of telomere biology in cell senescence” the authors should explain what deprotected telomeres are.

Author response: there are three sentences that contain “short or deprotected telomeres”. We have now removed the word “deprotected”, because the structures of deprotected telomeres are still unclear.  

The authors start the first paragraph of the review by indicating that telomeres cannot be fully replicated and discuss the activity of telomerase. How can the readers understand these concepts if the authors have not yet discussed the telomeric DNA sequence and structure? These first lines of the first paragraph should be placed at the end of the paragraph. The authors should at least mention that telomeres terminate with a G-rich 3’ overhang.

Author response: We have modified the paragraph by moving the first a couple of sentences to the end of the paragraph.

In several instances concepts are introduced but not explained. As result the review is difficult to read.
For example, at lines 119-122, the authors write “yeast silencing machinery, encoded by SIR2, SIR3, and SIR4, undergoes redistribution from telomeres to non-telomere sites when telomeres are shortened to increase the lifespan, and deletion of either SIR3 or SIR4 decreases the lifespan [47]” How would this work? Why redistribution of Sir proteins may increase lifespan? Is the telomeric chromatin involved? Is replicative stress at telomeres involved? Are extratelomeric roles of Sir proteins involved? Could telomere transcription be involved?

Author response: We have modified the sentence, so it now reads that “the yeast chromatin silencing machinery, encoded by SIR2, SIR3, and SIR4, undergoes redistribution from telomeres to non-telomere sites when telomeres are shortened to increase heterochromatin maintenance, genome stability and the lifespan, and deletion of either SIR3 or SIR4 decreases the lifespan [47].”

Similarly, at lines 223-225: “More intriguingly, the significantly increased genome instability is suppressed by down-regulation of the evolutionarily conserved TORC1 (the target of rapamycin complex 1)-Sch9 pathway [50].” Why is this intriguing? What does this indicate?

Author response: We have modified the sentence to readthe significantly increased genome instability is suppressed by down-regulation of the evolutionarily conserved TORC1 (the target of rapamycin complex 1)-Sch9 pathway [52], suggesting that pharmaceutical strategies (e.g. rapamycin and caffeine [83]) are applicable to stabilise the genome instability caused by constitutive active telomerase to increase lifespan.”

In the perspectives section, the authors write: “The mechanisms that determine the different aspects of longevity, premature replicative senescence or chronological ageing appear to hinge on the extensity and intensity of telomeric DNA abnormality, and on the fidelity of DNA damage response and repair”. Which telomeric DNA abnormalities that impact on senescence and longevity are the authors referring to? And which DNA repair pathways?

Author response: These questions cannot be answered for sure and are the areas that require further investigation. We have modified the sentence to read: “The mechanisms that determine the different aspects of longevity, premature replicative senescence or chronological ageing appear to hinge on the extensity and intensity of telomeric DNA abnormality, and on the fidelity of DNA damage response and repair which require further investigation”.

Still in the perspectives sections, what do the authors mean by “The different lengthening mechanisms are to different degrees at the expenses of risking premature ageing and related disorders”? or by “means to intercept stress-induced organismal premature ageing for longevity”?

Author response: We have modified the sentence to make it a little more concrete, so it now reads:“The different telomere lengthening mechanisms such as uncontrolled end-end joining are to different degrees at the expenses of risking premature ageing and related disorder such as cancer.”

The manuscript requires English editing:
some examples: "telomerase expresses in wild type yeast" line 65; "gradually loss telomere sequence" line 108

Author response: We have done proofreading and fixed the errors.